# The Efficacy of a Posterior Approach to Surgical Correction for Neglected Idiopathic Scoliosis: A Comparative Analysis According to Health-Related Quality of Life, Pulmonary Function, Back Pain and Sexual Function

**DOI:** 10.3390/children10020299

**Published:** 2023-02-03

**Authors:** Pawel Grabala, Ilkka J. Helenius, Jacob M. Buchowski, Suken A. Shah

**Affiliations:** 1University Children’s Hospital, Department of Pediatric Orthopedic Surgery and Traumatology, Medical University of Bialystok, Waszyngtona 17, 15-274 Bialystok, Poland; 2Paley European Institute, Al. Rzeczypospolitej 1, 02-972 Warsaw, Poland; 3Department of Orthopedics and Traumatology, Helsinki University Hospital, 00260 Helsinki, Finland; 4Department of Orthopedic Surgery, Washington University School of Medicine, 660 S Euclid Ave, St. Louis, MO 63110, USA; 5Department of Orthopaedic Surgery, Nemours Alfred I. duPont Hospital for Children, 1600 Rockland Road, Wilmington, DE 19803, USA

**Keywords:** severe scoliosis, HRQoL, pulmonary function, scoliosis surgery

## Abstract

Background: This study aimed to evaluate the treatment outcomes of severe idiopathic scoliosis (IS) and hypothesized that surgical treatment would have a superior impact on the health-related quality of life (HRQoL), pulmonary function (PF), back pain, and sexual function. Methods: We retrospectively reviewed 195 consecutive patients with IS classified into severe (SG) and moderate groups (MG) with a minimum follow-up of two years. Results: The mean preoperative curve was 131° and 60° in the SG and MG, respectively. The mean preoperative flexibility in the bending films averaged between 22% in the SG and 41% in the MG. After definitive surgery, the main curve was corrected to 61° and 18° in the SG and MG, respectively. The mean preoperative thoracic kyphosis was 83° in the SG and 25° in the MG, which was corrected to 35° in the SG and 25° in the MG. At baseline, the percentage of predicted lung volume (FVC) was significantly lower in the SG than that in the MG (51.2% vs. 83%). The baseline percentage of the predicted FEV1 values was also significantly lower in the SG than in the MG (60.8% vs. 77%). During the two-year follow-up, the percentage of predicted FVC showed significant improvement in the SG (69.9%) (*p* < 0.001), and the percentage of predicted FEV1 values during the follow-up improved significantly in the SG (76.9%) (*p* < 0.001) compared with the MG (81%), with no statistical difference observed during the two-year follow-up. The SRS-22r showed a clinically and statistically significant improvement in the preoperative results to those of the final follow-up (*p* < 0.001). Conclusions: Surgical treatment of severe scoliosis can be safe. It provided a mean correction of the deformity for 59% of patients and significantly improved respiratory function, with the percentage of predicted forced expiratory volume in 1 s improving by 60% and the forced vital capacity improving by 50%, resulting in clinically and statistically significant improvements in the SRS-22r, HRQoL outcome scores, and back pain (reduced from 36% to 8%), as well as improved sexual function. The planned surgical treatment can achieve a very significant deformity correction with a minimal risk of complications. The surgical treatment has a superior impact on the quality of life patients with severe spinal deformities and significantly improves function in every sphere of life.

## 1. Introduction

Most spinal deformities in children and adolescents are of unknown etiology and are therefore termed idiopathic [1]. While the treatment of curvatures of up to 90° is not a major problem for an experienced surgical team, the management of spinal deformities in children and adolescents with curvatures over 90°, which are sometimes stiff, is a challenge even for modern practitioners of medicine [2]. Severe scoliosis includes deformities with a Cobb angle ≥ 90°, with flexibility less than 30% [3], and is extremely difficult to correct in a one-stage surgery [4,5,6,7,8].

The usual operating strategies used to deal with the severity and stiffness of spinal deformities include HGT and halo-pelvis (HP) or halo-femoral traction (HFT) as safe preoperative adjunct devices, which gradually straighten the spine prior to spinal surgery, in order to obtain the greatest possible correction with a minimal risk of spinal cord injury [6,7,9,10,11,12,13]. Experienced surgeons can also use alternative techniques, such as temporary internal distraction rods (TID), intraoperative traction (IT), osteotomies, and apical spine resection, in combination with anterior and posterior approaches to achieve safe and optimal results [6,7,9,10,11,12,13]. Unfortunately, there are limited data in the literature describing the quality and improvement in the life of patients who undergo surgery for massive and neglected spinal curvatures [14,15,16,17]. There are many studies evaluating the results of surgical treatment, quality of life, pain, sexual function, degree of disability after surgery of the lumbar spine, or treatment of moderate scoliosis. However, there are very few studies showing the impact of surgical correction on the quality of life of patients with severe scoliosis and comparing them to the outcomes of the management of scoliosis with a smaller curve [18,19,20,21,22,23,24]. The aim of the present study was to compare two groups of patients with severe and moderate spine deformities to better delineate the differences in back pain, HRQoL, sexual satisfaction or dysfunction, and PF in patients with severe idiopathic scoliosis (IS) who underwent surgical treatment involving pedicle screw instrumentation (staged, intraoperative traction, or instrumentation with HGT), compared to patients with moderate typical IS who underwent one-stage posterior spinal fusion (PSF), with a minimum postoperative follow-up of two years. In this study we can more thoroughly analyze the assessed parameters in the study groups. We hypothesized that the surgical treatment of severe scoliosis (traction or staged) would result in reduced spinal deformity, back pain, and improved HRQoL, sexual function (SF), and PF, similar to posterior spinal fusion (PSF) for typical, moderate IS curves.

## 2. Patients and Methods

### 2.1. Setting and Patients

There were selected and included in the retrospective study 88 adolescents with severe scoliosis deformities (SSDs) who were treated with preoperative HGT (28), intraoperative traction (25), or staged surgery, with temporary rods and internal distraction (35) followed by PSF to evaluate the outcomes of the treatment in connection with spinal deformity correction, PF, HRQoL, and SF (SG group). These adolescents were matched by age and sex with 107 patients with typical thoracic idiopathic scoliosis (major curve (MC), 50°–70°) to form the control group (MG group). All patients underwent surgical treatment in the years 2016–2020 by the same senior orthopedic spine surgeon (the first author) and were followed up for a minimum of two years (Figure 1).

The inclusion criteria were as follows: those with severe (juvenile or adolescent) idiopathic scoliosis (major curve Cobb ≥ 90° with a flexibility < 30%) were classified as the severe scoliosis group (SG), and those with moderate thoracic idiopathic scoliosis (MC, 50°–70°) treated with PSF in a one-stage procedure were classified as the moderate scoliosis group (MG). All patients underwent PSF with segmental pedicle screw implantation using well-known correction maneuvers from the literature [25,26]. The surgical procedure for severe scoliosis used HGT, TID, or IT followed by standard pedicle screw placement and three-dimensional correction. HGT and TID were performed as an initial surgery, and the final correction was performed as the second stage [25,26]. The HGT as described in the literature [2,27,28,29,30,31] was used from 4 to 6 weeks, until reaching 45% of the patient’s body weight in the traction. TID rods were used as described in the literature by Buchowski and Skaggs [9,10,11] or with our own modification [32]. The surgical technique for IT was performed as described in the literature [33]. All surgical procedures were performed by the senior orthopedic surgeon (the first author) under neuromonitoring control. The study was approved by the ethics committee of the hospital district.

### 2.2. Outcome Parameters

The evaluated parameters were as follows: rib hump, trunk height, radiographic outcomes, PF tests (forced vital capacity (FVC) and forced expiratory volume in 1 s (FEV1) as a percentage of the predicted values), perioperative complications, HRQoL using the SRS-22r questionnaire [15,34], sexual function using the FSDS (Female Sexual Distress Scale) only for female patients [35], visual analogue scale (VAS) score, Oswestry Disability Index (ODI), and the EQ-5D questionnaires [36,37]. We collected all parameters preoperatively, after the final fusion, and during the two-year follow-up. All complications, if they had appeared, were also noted.

### 2.3. Radiographic Parameters

Standard standing posteroanterior and lateral radiographs of the entire spine were obtained preoperatively, after surgery, and during the final follow-up. Preoperative bending radiographs were obtained to evaluate the flexibility of the curves. The curves were classified according to the Lenke classification [26]. The Cobb angles of the proximal thoracic, main thoracic, and lumbar curves were noted, and sagittal measurements—thoracic kyphosis (T5–T12), lumbar lordosis (T12–S1)—were also included. Additionally, the apical vertebral translation (AVR) was measured. The thoracic rib hump was measured preoperatively, postoperatively, and at the final postoperative follow-up. The correction percentages of the main curves were then calculated. The trunk height noted between the lowest instrumented vertebra and T1 was measured between the midpoint of the upper T1 endplate and the midpoint of the lower endplate of the lowest instrumented vertebra. The radiographic measurements were performed by an independent observer. All treated patients had ordered an MRI for excluding other spinal cord pathology. The PF was examined via the standard protocol using a spirometer in a sitting position, preoperatively and at the final follow-up. We noted the PF values as the following: FVC and FEV1, expressed as a percentage of the predicted values (%). All patients underwent intraoperative spinal cord monitoring, including somatosensory evoked potentials (SSEP) and transcranial motor-evoked potentials (MEP) [10].

### 2.4. Statistical Analysis

In our study, we used statistical analysis software (version 10.0; StatSoft Inc., Tulsa, OK, USA) for all analyses. ANOVA and the Tukey–Kramer method were used. For the analysis and presentation of the data, we used standard deviation (SD) as the means, 95% confidence interval (CI), or as medians with lower and upper quartiles or frequency, as appropriate. The normal distribution assumption was checked visually together with a Shapiro–Wilk test. The Mann–Whitney *U*-test and Kruskal–Wallis analysis of variance rank test were used for between-group comparisons. Pearson’s correlation coefficients were calculated to examine the association between the two numerical variables. Changes between the two time points were compared using McNemar tests. A *p*-value of <0.05 was considered statistically significant.

## 3. Results

### 3.1. Clinical Characteristics, Radiographic and PF Outcomes

In this study, we enrolled 78 girls and 10 boys (SG) with severe idiopathic scoliosis with a mean (SD) age of 14.3 (2.8) years, who were treated with preoperative HGT (28), intraoperative traction (25), or staged surgery, with temporary rods and internal distraction (35) followed by PSF, and 93 girls and 14 boys with a mean (SD) age of 15 (2.6) years (*p* = 0.921) were enrolled in the MG (Table 1).

The mean (SD) trunk height increased from 31.8 (2.8) cm to 41.8 (3.2) cm in the SG, and from 29.2 (4.2) cm to 38 (3.6) cm in the MG (*p* < 0.001 for comparison between the SG and MG; Figure 2, Figure 3, Figure 4 and Figure 5).

The mean (SD) preoperative MC was 131° (13.8) and 60° (8.4) in the SG and MG, respectively (*p* < 0.001). The mean (SD) flexibility of curves before surgical treatment measured in the bending films were noted as 22% (7.2) in the SG and 41% (15.5) in the MG (*p* < 0.001). After definitive surgery, the MC was corrected to 61° (19.2) and 18° (9.2) in the SG and MG, respectively (*p* < 0.001). The mean percentage correction of the MC was similar in both the groups (59% vs. 65% in the SG and MG, respectively), with no statistically significant difference between the groups (NS). No progression of MC was observed during the follow-up (Table 1). There was no noted statistical difference in the final correction rate in the coronal plane between patients managed with HGT and TID, but there was observed a slightly smaller correction between HGT, TID patients (*p* < 0.001) and only IT patients (*p* < 0.001).

The mean preoperative thoracic kyphosis was 83° (35.9) in the SG and 25° (14.2) in the MG (*p* < 0.001). It was corrected to 35° (9.6) in the SG and 22° (8.2) in the MG (*p* < 0.001). The mean preoperative lumbar lordosis was −66.1° (10.8) in the SG and −52° (11) in the MG (*p* = 0.329), which was corrected to −16.8° (8.8) in the SG and −18° (12) in the MG (*p* = 0.897). The mean preoperative apical vertebral translation improved from 9.2 cm (3.4) to 3.1 cm (0.98) at the final follow-up in the SG and from 6.5 cm (1.48) to 2.2 cm (0.68) at the final follow-up in the MG (*p* < 0.001). There was no noted statistical difference of the final correction rate in the sagittal plane and AVT between patients managed with HGT and TID, but there was observed a significant improvement in the HGT and TID patients than in IT patients (*p* < 0.001).

At baseline, the percentage of predicted lung volume (FVC) was significantly lower in the SG than in the MG (51.2% vs. 83%; *p* < 0.001). The baseline percentage of the predicted FEV1 values was also significantly lower in the SG than in the MG (60.8% vs. 77%; *p* < 0.001). During the two-year follow-up, the percentage of the predicted FVC showed significant improvement in the SG (*p* < 0.001), but not in the MG (mean, 69.9% vs. 79%; NS). Likewise, the percentage of the predicted FEV1 values during the follow-up improved significantly in the SG (*p* < 0.001) compared with the MG (mean, 76.9% vs. 81%), with no statistical difference observed during the two-year follow-up (Table 1).

### 3.2. HRQoL

We noted a significant improvement in the mean preoperative SRS-22r total score from 2.96 to 4.36 in the SG and from 3.82 to 4.26 in the MG during the follow-up period (*p* < 0.001 for both comparisons) (Table 2). The SRS-22r showed a clinically and statistically significant improvement from the preoperative results to those of the final follow-up (*p* < 0.001).

The mean preoperative EQ-5D total score improved significantly from 56 to 73 in the SG (*p* < 0.001), and non-significantly from 73 to 78 in the MG (NS). The mean preoperative pain VAS score improved significantly from 5.9 to 2.9 and 5.15 to 3.1 for both the SG and the MG (*p* < 0.001). The mean preoperative ODI improved significantly from 36.4 to 10.4 and from 31.4 to 8.5 for both the SG and the MG (*p* < 0.001). The mean preoperative FSDS total score improved significantly from 13.6 to 6.8 in the SG (*p* < 0.001) and non-significantly from 10.2 to 4.5 in the MG (NS) (Table 3).

### 3.3. Complications

In the study groups we have found that 19% of the patients in the SG and 22% of the patients in the MG had experienced postoperative complications. None of the patients received a new postoperative neurological deficit in the SG and MG (Table 4). No complications were reported during the final follow-up.

## 4. Discussion

In this study, we proved that the surgical treatment of severe scoliosis can be safe when the most adequate and optimal method is chosen. We concluded that the surgical treatment used can ensure better HRQoL and PF, which can be obtained using a combination of preoperative HGT, intraoperative traction, or staged surgical treatment with temporary internal distraction, followed by segmental screw placement and spinal fusion [2,9,10,11,38]. The above procedure can achieve good correction with a decreasing risk of permanent neurological deficits or PF (Table 4) compared with the literature (Table 5).

### 4.1. Correction of Spinal Deformity

Our study showed a mean correction of 59% without the use of radical surgical technique methods (anterior surgery or VCR); however, our surgical procedure assumed multiple posterior column osteotomies as SPO, Ponte’s osteotomies at several levels on the apical levels of the thoracic and lower lumbar curves. There are some reports in the literature of receiving a significant correction using a combination of several surgical techniques and approaches such as the anterior release, Halo traction, and finally PSF after undergoing very aggressive release and a longer treatment period in the hospital of more than three weeks. Jasiewicz et al. received only a 44% correction using a combined approach (anterior release and PSF) [40]. Other studies by Potaczek et al. showed a mean correction of 53% using the anterior approach and HFT followed by PSF for severe scoliosis [41]. In their study it was also noted that there was a high incidence of complications (10.5%). Lenke et al. received a mean decrease in 51% of the main curve for severe pediatric spinal deformities, including surgical treatment with the VCR technique [39]. VCR corrects multiplanar deformities, including the resection of one or more spinal segments. In the adult scoliosis group, an MC > 100° with a flexibility of 18.2% showed an immediate main curve correction of 56.4% [39,42]. Our patients had a complication rate of 19% in the SG group and 22% in the MG group (NS), as shown in Table 4. No permanent neurological deficit or decreased respiratory function was observed. It is already known that other aggressive procedures can cause more pulmonary complications [6,29,44]. Preoperative techniques include halo gravity traction, halo femoral traction, or intraoperative traction that are well described in the literature and are also widely used as safe and effective techniques for the management of severe spinal deformities before final surgical correction and fusion. They allow for a partial, less invasive, and safer correction of major and stiff curves, often with compensatory curves, so that the final correction and fusion with transpedicular screws can be performed on a less severe and rigid curve [4,27,28,29,30,31]. There are some long-term studies which explain that spinal balance and corrections can be still stable without any decompensation over time, and MRI studies have demonstrated similar disc and facet degeneration rates for the L3 and L4 groups [45]. The clinical reports showed similar outcomes for all groups at the 5-year observation period. In another study by Akazawa et al., AIS patients at a 35-year follow-up after spinal fusion surgery showed changes in the lumbar spine such as disc degeneration and Modic changes in the non-fused segments. Additionally, reduced lumbar lordosis, SVA imbalance, and severe disc degeneration were observed in these patients compared to the fusion levels of L4, L3, or higher [46].

### 4.2. Back Pain

In this study, back pain (BP) before surgery was noted in 36% of patients in the SG group and was rated as moderate to severe. A considerable percentage (28%) of our patients in the MG group preoperatively rated BP as moderate to severe, which is similar to other studies by Ramirez et al., who noted that 23% of patients with AIS were noted with BP at the time of diagnosis [47], and Sieberg et al., who showed that, of 190 patients with AIS, 35% reported moderate to severe pain preoperatively [48]. Some studies have described the prevalence of BP in AIS patients before surgical treatment as being from 42% to 80% [23,48,49,50,51,52], but the prevalence of BP in patients with AIS and its relation to spinal deformity remains unclear [14,15,17,52]. In our study, there was no noted statistical difference in the back pain between patients managed with HGT, TID, and IT (*p* < 0.001). In the study of Grabala et al., the authors have noted a 34% prevalence in the back pain in patients who had experienced the surgical treatment of scoliosis compared to healthy controls [53]. In the present study, BP after surgery was reported by only 8% of the patients during the final follow-up. The mean pain domain score improved from 3.22 to 3.98 for the SG and from 3.92 to 3.98 for the MG group, and the VAS score improved from 5.9 to 2.9 for the SG and from 5.15 to 3.1 for the MG during the final follow-up. These data are similar to those reported by Helenius et al. [24]. There were no differences between the spinal fusion and L3 or L4 [53]. Additional long-term follow-up data are needed in regard to this topic.

### 4.3. HRQoL, PF, and SF

Analyzing PF in the study groups, we noted that patients with SSDs have been in general, severely impaired, which can cause a higher risk of scoliosis correction surgery and affect the daily activity, growth, development, and appearance of the patient [28,54]. The literature describes a strong correlation between the curve correction and improvement in PF (Table 5). Improving respiratory function before surgery for elective scoliosis correction may reduce the risk of postoperative complications [14,29]. In our study, we arrived at similar conclusions. The surgical treatment of severe scoliosis with halo devices or staged surgery significantly decreases the translation of the apex deformity by 70% and improves postoperative PF in patients with severe scoliosis [9,10]. Restrictive pulmonary disease is frequently observed in patients with severe spinal deformities and may lead to increased morbidity and mortality [28]. Bumpass et al. concluded that the pediatric patients they studied who underwent PVCR had a small significant mean increase in FVC and FEV1 at the two-year follow-up [55]. In their study, the mean percentage of the predicted FVC and FEV1 decreased slightly but not significantly. A 3% decrease was noted in the predicted FVC, and a 1% decrease was noted in the predicted FEV1 [55]. In the present study, the mean preoperative FVC predicted values (%) were significantly worse in the SG than in the MG. At the two-year follow-up, the FVC and FEV1 values showed a statistically significant improvement in the severe scoliosis group but not in the moderate IS group. These findings imply that traction or staged surgery is advantageous for PF in this cohort. More importantly, the staged surgery for SSD resulted in significant improvements in both the disease-specific (SRS-22r) and general HRQoL (EQ-5D) patients during the two-year follow-up. Some studies have evaluated the incidence of pulmonary complications, with their preoperative FEV1 being an independent predictor of pulmonary complications. The observed mortality rate due to pulmonary complications in the treatment of complex spinal deformities has demonstrated the urgent need for the careful analysis and preoperative preparation of high-risk patients to minimize complications. [28,55,56]. In our study, SF significantly improved in the SG and without a statistically significant difference in the MG. Surgical treatment serves to improve the quality of life in many aspects of functioning. It is associated with improving the appearance of and reducing or removing body deformities of patients, as shown in the study. Some researchers have shown [57] that patients with AIS who underwent the surgical correction of scoliosis may suffer from sexual disorders and dysfunctions, with reduced sexual satisfaction, less frequent orgasms, and reduced sexual arousal, even many years after the end of the treatment. In another study [53], no difference or sexual dysfunction was noted in AIS patients who underwent PSF in comparison to healthy controls. In a study of sexual dysfunction/satisfaction after spinal surgery, Daniels et al. showed that the score of sexual dysfunctions caused by lumbar stiffness significantly improved after surgical treatment of the spinal deformity. In addition, the sexual dysfunction associated with lumbar stiffness was reported to be strongly associated with the overall score, as measured by the ODI and SRS-22r, and sexual satisfaction was closely related to the increase in the quality of life after surgery [58].

A five-year follow-up study on VCR outcomes showed a significant improvement in the SRS-24/22 outcome questionnaire but did not report the data on PF [59]. Extensive surgical procedures, operative time, and surgical site infection are additional concerns. In our study, all patients recovered after surgery with no major complications.

### 4.4. Limitations

A strength of this study is the post-surgical observational period. All patients participating in the study were treated, considered for qualification for surgical treatment, and operated on by the same experienced surgeon. All patients were under constant observation after surgery for at least 2 years, and we were able to provide both full pre- and postoperative data, as well as data on HRQoL, sexual function, back pain, ODI, PF, and SRS-22r using standard performance measurements. This study was limited by its retrospective nature and relatively small sample size. It is very difficult to obtain two groups with severe deformities that can be treated by other surgical techniques that can be compared with one another. Fortunately, the epidemiology of severe spinal deformities is small in relation to moderate idiopathic scoliosis. Accordingly, we compared the treatment of severe scoliosis with that of moderate scoliosis.

## 5. Conclusions

The surgical management of neglected and severe spinal deformities has a major impact on the quality of life of patients and significantly improves functioning in every sphere of life, with a limited number of complications. The surgical procedures can be safe, provided a mean correction of deformity of 59%, and significantly improved respiratory function on average by 50%, resulting in clinically and statistically significant improvements in the SRS-22r, HRQoL, ODI, VAS, EQ-5D, FSDS outcome scores, improved sexual function, and reduced back pain.

## Figures and Tables

**Figure 1 children-10-00299-f001:**
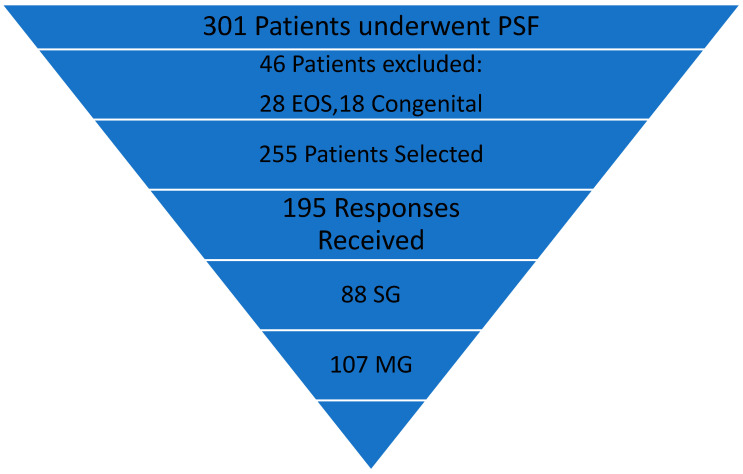
A flowchart showing identified patients with idiopathic scoliosis treated with spinal fusion selected for our study.

**Figure 2 children-10-00299-f002:**
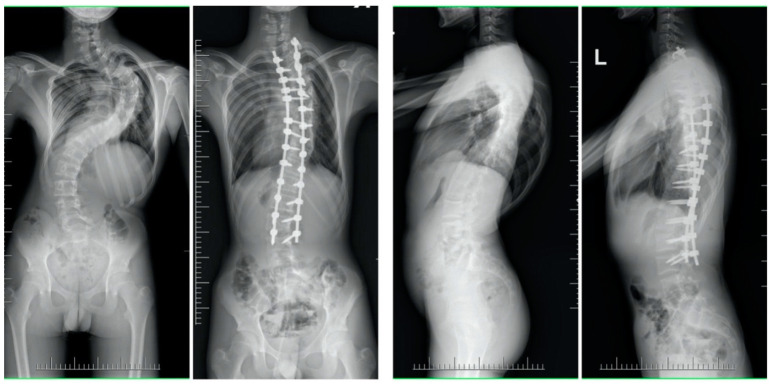
Radiographs of a 17-year-old patient with a severe spinal deformity before surgery and at the final follow-up visit.

**Figure 3 children-10-00299-f003:**
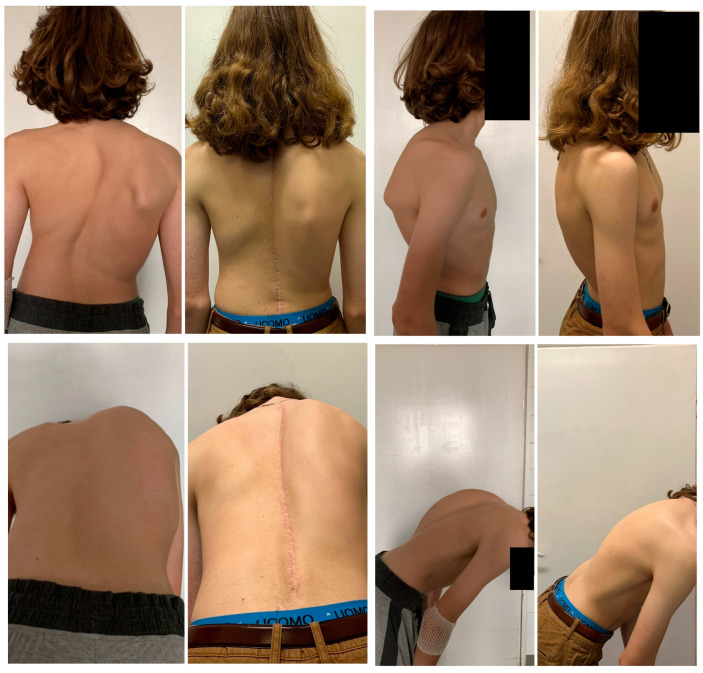
The clinical pictures of a 17-year-old patient with a severe spinal deformity before surgery and at the final follow-up visit.

**Figure 4 children-10-00299-f004:**
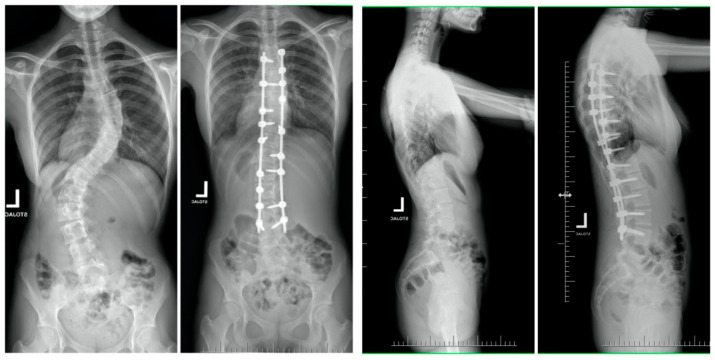
Radiographs of a 15-year-old patient before the one-staged surgery and at the final follow-up visit.

**Figure 5 children-10-00299-f005:**
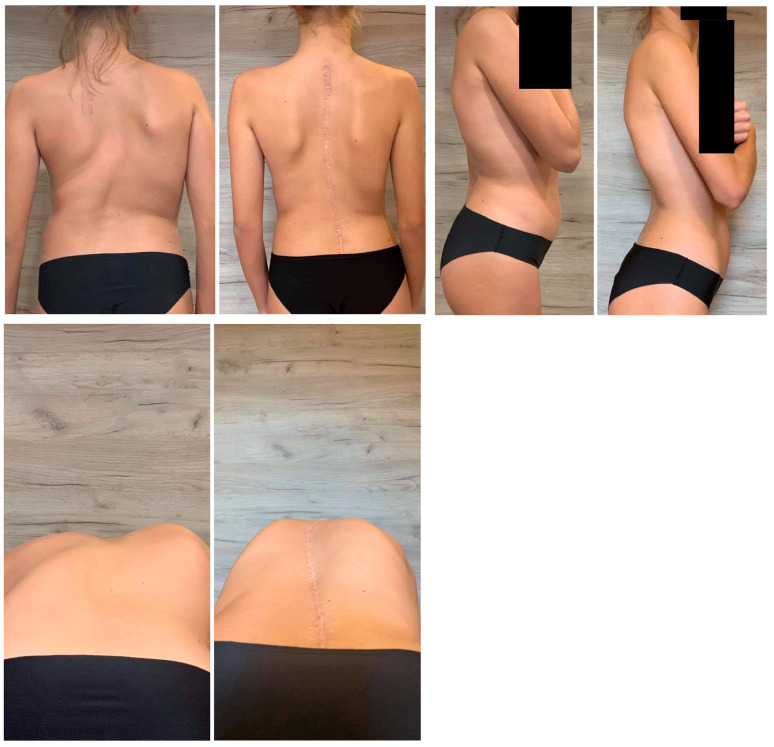
The clinical pictures a 15-year-old patient before the one-staged surgery and at the final follow-up visit.

**Table 1 children-10-00299-t001:** The clinical characteristics of the study groups.

	SG (*N* = 88)	MG (*N* = 107)	*p* Value *
Mean (SD) age at surgery, yrs	14.3 (2.8)	15 (2.6)	0.921
Male	10	14	
Female	78	93	
Mean (SD) follow-up, yrs	4 (3.5)	4 (2.5)	0.922
Mean (SD) BMI, kg/m^2^	19.2 (2.4)	18.7 (2.2)	0.671
Mean (SD) preoperative Cobb, deg	131 (13.8)	60 (9.5)	<0.001
Mean (SD) Cobb at final follow-up, deg *p*-value * (pre vs. follow-up)	61 (19.2)	18 (9.2)	<0.001
	<0.001	<0.001	
Mean (SD) preoperative thoracic kyphosis T5-T12, deg	83 (35.9)	25 (14.2)	<0.001
Mean (SD) thoracic kyphosis T5-T12 at final follow-up, deg *p*-value * (pre vs. follow-up)	35 (9.6)	22 (8.2)	0.24
	<0.001	<0.324	
Mean (SD) preoperative lumbar lordosis T12-S1, deg	−66.1 (10.8)	−52 (11)	0.329
Mean (SD) lumbar lordosis T12-S1 at final follow-up, deg *p*-value * (pre vs. follow-up)	−16.8 (8.8)	−18 (12)	0.897
	<0.001	<0.001	
Mean (SD) preoperative apical vertebral translation, mm	92 (32.4)	65 (14.8)	<0.001
Mean (SD) apical vertebral translation at final follow-up, mm *p*-value * (pre vs. follow-up)	31 (9.8)	22 (6.8)	0.22
	<0.001	<0.001	
Mean (SD) preoperative forced vital capacity, percentage of predicted	51.2 (12.8)	83 (11.2)	<0.001
Mean (SD) forced vital capacity, percentage of predicted at final follow-up *p*-value * (pre vs. follow-up)	69.9 (11.2)	79 (13.2)	0.486
	<0.001	0.12	
Mean (SD) preoperative forced expiratory volume in one second, percentage of predicted	60.8 (13.9)	77 (12.8)	<0.001
Mean (SD) forced expiratory volume in one second, percentage of predicted at final follow-up *p*-value * (pre vs. follow-up)	76.9 (14.5)	81 (12.8)	0.282
	<0.001	0.09	
Mean (SD) preoperative rib hump difference, cm	8.6 (2.4)	4.2 (1.9)	<0.001
Mean (SD) rib hump difference, cm, at final follow-up *p*-value * (pre vs. follow-up)	2.4 (1.8)	1.61 (2.1)	0.682
	<0.001	<0.001	
Mean (SD) preoperative trunk height difference, cm	31.8 (2.8)	29.2 (4.2)	<0.001
Mean (SD) trunk height difference, cm, at final follow-up *p*-value * (pre vs. follow-up)	41.8 (3.2)	38 (3.6)	0.472
	<0.001	<0.001	
Mean (SD) number of levels fused	11 (4.0)	9 (3.0)	0.964
Percentage (n) of patients fused below L3	41% (41)	NA	0.543
Halo Gravity Traction (n)	28	NA	
Halo Gravity Traction, Mean (SD) days of traction	42 (8)	NA	
Temporary Internal Distraction (n)	35	NA	
Temporary Internal Distraction Mean (SD) time between initial and final surgery, days	28 (7)	NA	
Intraoperative Traction (n)	25	NA	

* 2-sided *t* test or Wilcoxon test.

**Table 2 children-10-00299-t002:** Scores of the SRS-22R in the surgical study groups.

SRS-22R	SG (*N* = 88)			MG (*N* = 107)		
Parameter	Preoperative (*N* = 88)	Final Follow-Up (*N* = 88)	*p*-Values *	Preoperative (*N* = 107)	Final Follow-Up (*N* = 107)	*p*-Values *
Function	2.80 (0.82)	4.42 (0.66)	<0.001	4.10 (0.52)	4.32 (0.52)	0.309
Pain	3.22 (0.76)	3.98 (0.78)	<0.001	3.92 (0.55)	3.98 (0.60)	0.268
Self-image	2.86 (0.76)	4.12 (0.66)	<0.001	3.86 (0.52)	3.96 (0.72)	0.421
Mental health	2.68 (0.72)	4.02 (0.70)	<0.001	4.02 (0.72)	4.12 (0.70)	0.629
Satisfaction	2.60 (0.80)	4.30 (0.60)	<0.001	3.80 (0.76)	4.22 (0.70)	<0.001
Total score	2.96 (0.82)	4.36 (0.55)	<0.001	3.82 (0.82)	4.26 (0.75)	<0.001
*p*-Values *	<0.001					

* Statistical comparisons were performed using the Kruskal-Wallis test; *p* < 0.05 for all.

**Table 3 children-10-00299-t003:** HRQoL in the surgical study groups. Values are mean (SD).

Parameter (Mean)	SG (*N* = 88)	MG (*N* = 107)	*p*-Value
EQ-5D			
Preoperative	56 (18)	73 (18)	<0.001
At FFU	73 (23)	78 (20)	0.391
*p*-Value	<0.001	0.12	
VAS score			
Preoperative	5.9 (2.2)	5.15 (2.5)	<0.001
At FFU	2.9 (2.0)	3.1 (2.2)	<0.001
*p*-Value	<0.001	<0.001	
ODI			
Preoperative	36.4 (18)	31.4 (16)	<0.001
At FFU	10.4 (8)	8.5 (6.2)	0.29
*p*-Value	<0.001	<0.001	
FSDS			
Preoperative	13.6 (3.8)	10.2 (2.6)	<0.001
At FFU	6.8 (3.2)	4.5 (3.8)	0.621
*p*-Value	<0.001	0.22	
SRS-22R (total)			
Preoperative	2.96 (0.82)	3.82 (0.82)	<0.001
At FFU	4.36 (0.55)	4.26 (0.75)	<0.001
*p*-Value	<0.001	<0.001	

**Table 4 children-10-00299-t004:** The rate of complications following the posterior final fusion.

Rate of Complications Following Posterior Final Fusion	SG (*N* = 88)	MG (*N* = 107)
Intraoperative neuromonitoring changes	5 (5.68%)	6 (5.6%)
Superficial infection	3 (3.40%)	5 (4.67%)
Pneumonia	2 (2.27%)	3 (2.80%)
Paresthesia from the lateral cutaneous nerve of the lower limb	1 (1.13%)	6 (5.6%)
Superior mesenteric artery syndrome	2 (2.27%)	0
Deep infection/revision surgery	2 (2.27%)	1 (0.93%)
Radiculopathy	1 (1.13%)	2 (1.86%)
Implant failure (broken rod/screws/pull out)	2 (2.27%)	1 (0.93%)
TOTAL	17 (19.31%)	24 (22.42%)

**Table 5 children-10-00299-t005:** The studies showing outcomes of severe scoliosis correction.

Authors	Surgical Technique	Mean Coronal Cobb (Degree)	Mean Cobb Correction (%)
Yu Wang et al. [4]	Halo pelvic traction/posterior fusion	131.5	64.1
Yu Wang et al. [4]	Halo pelvic traction/posterior column resection	133.6	65.4
Skaggs et al. [10]	Temporary internal distraction rods	113	54
Hui-Min Hu et al. [11]	Temporary internal Distraction rods	148.8	63
Zhang et al. [12]	Posterior fusion/Ponte osteotomy	98.5	56.7
Zhang et al. [12]	Posterior vertebral column resection	108.9	49.2
Koller et al. [13]	Temporary treatment with magnetically controlled growing rod for surgical correction of severe adolescent idiopathic thoracic scoliosis greater than 100°.	118	67
Wang et al. [25]	VCR	108.9	66.1
Rinella et al. [27]	Halo-gravity traction	84	59
Nemani et al. [28]	Halo-gravity traction	131	56
Hamzaoglu et al. [33]	Halo-gravity traction and PSF	122	51
Lenke et al. [39]	VCR	85	51
Jasiewicz et al. [40]	Anterior release, cranio-femoral traction, and PSF	129	44
Potaczek T et al. [41]	Halo-femoral traction with PSF, anterior release with halo-femoral traction, and PSF	125	52.7 and 51.7
Lenke et al. [42]	Posterior vertebral column resection	85	69
Di Silvestre et al. [43]	Severe adolescent idiopathic scoliosis: posterior staged correction using a temporary magnetically controlled growing rod.	98.2	59

## Data Availability

Not applicable.

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
