# Peer review of "The Efficacy of a Posterior Approach to Surgical Correction for Neglected Idiopathic Scoliosis: A Comparative Analysis According to Health-Related Quality of Life, Pulmonary Function, Back Pain and Sexual Function"

_children, 2023, doi:10.3390/children10020299_

Round 1

Reviewer 1 Report

I congratulate the authors  - 2 years minimum follow-up and good data relating to SRS22 and lung function. Most other studies have looked at lung function in children with neurological disorders with minimal/no evidence of improvement with surgery, often attributed to the underlying neurological condition. Here we have a study group with idiopathic curves.

I have 2 concerns which should be addressed:

The authors state "88 consecutive adolescents with severe scoliosis 72 deformities (SSDs) who were treated with preoperative HGT, intraoperative traction, or 73 staged surgery, with temporary rods and internal distraction followed by PSF ..."  - the number treated by each method should be stated, and although the numbers may not be sufficient for statistical analysis, some commentary regarding duration and relevant effectiveness of each method is warranted.

I note authors from Finland and USA. Was this a combined study from 2 countries or single source? There is some inconsistency noting "All procedures were 86 performed by the senior orthopedic surgeon" in Methods and later "All patients participating in the study were treated, considered for qualification for surgical treatment, and 295 operated on by the same experienced surgeons using similar surgical techniques" in Limitations. This suggest a combined series and this should be defined in Methods.

Was it 86 or 88 procedures/patients, noting above?

Author Response

Dear Sir,

Thank you for your important review and great comments.

I agree with you. I have fixed all your suggestions.

I put number of patients treated with Halo Gravity Traction, Temporary Internal Distraction Rods, and with Intraoperative traction. I also marked, that all patients were treated by only one surgeon (the first author). Of course this work cannot be done without hard work of co-authors.

I put also the flowchart showing selecting patients to our study. I corrected several typos which I have found myself. I sorted the references.

Thank you for your review.

Best wishes,

Paweł Grabala

Reviewer 2 Report

1. Please mention study design in the title

2. Abstract: Well written

3. Intro: Purpose of the study needs to be clearer. What is lacking in the current literature on this subject? What is achieved additionally through this study?

4. Methods: Were these consecutively treated patients? How many patients with severe and moderate scoliosis were totally treated at the centers during the study period?

Table 1 needs to be included in the results section

Please briefly describe the techniques used in severe scoliosis (relevant details for length of HGT, additional maneuvres and osteotomies)

Please explain the definition of severe scoliosis

Results: Was any subgroup analysis possible to determine if any pre- or intra-operative variable influenced the outcome with regard to HRQOL, pain or pulmonary function

Please discuss relevant literature better for the functional and quality of life indicators of 

Discussion: Overall well discussed. Limitations and conclusions can be shortened and made clearer

Author Response

Dear Sir,

Thank you for your important review and great comments.

I agree with you. I have fixed all your suggestions as listed below:

  1. Please mention study design in the titleI corrected it.

2. Abstract: Well written

Thank you.

3. Intro: Purpose of the study needs to be clearer. What is lacking in the current literature on this subject? What is achieved additionally through this study?

I rebuild it. Thank you for your suggestion. 

4. Methods: Were these consecutively treated patients? How many patients with severe and moderate scoliosis were totally treated at the centers during the study period?

I put number of patients treated with Halo Gravity Traction, Temporary Internal Distraction Rods, and with Intraoperative traction. I also marked, that all patients were treated by only one surgeon (the first author).

I put also the flowchart showing selecting patients to our study. I corrected several typos which I have found myself.

Thank you for your suggestions.

Table 1 needs to be included in the results section

I fixed it. Thank you.

Please briefly describe the techniques used in severe scoliosis (relevant details for length of HGT, additional maneuvres and osteotomies)

I put it on text and add references for surgical techniques. The article is very extensive, contains a lot of data, so more data and tables can be tiring and confusing.

Please explain the definition of severe scoliosis

I fixed it.

Results: Was any subgroup analysis possible to determine if any pre- or intra-operative variable influenced the outcome with regard to HRQOL, pain or pulmonary function

I add the comments in the text. 

Please discuss relevant literature better for the functional and quality of life indicators of 

Discussion: Overall well discussed. Limitations and conclusions can be shortened and made clearer

I agree with you. Thank you. 

I corrected several typos which I have found myself. I corrected discussion and conclusion (a little shorter). I add some new references and sorted all list the references.

Thank you for your review. It was very important.

Best wishes,

Paweł Grabala

Round 2

Reviewer 2 Report

The recommended changes have been added.  The manuscript may be accepted in its current form.